# Cytotoxicity of Ficus Crocata Extract on Cervical Cancer Cells and Protective Effect against Hydrogen Peroxide-Induced Oxidative Stress in HaCaT Non-Tumor Cells

**DOI:** 10.3390/plants10010183

**Published:** 2021-01-19

**Authors:** Brenda De la Cruz-Concepción, Mónica Espinoza-Rojo, Patricia Álvarez-Fitz, Berenice Illades-Aguiar, Macdiel Acevedo-Quiroz, Ana E. Zacapala-Gómez, Napoleón Navarro-Tito, Hilda Jiménez-Wences, Francisco I. Torres-Rojas, Miguel A. Mendoza-Catalán

**Affiliations:** 1Laboratorio de Biomedicina Molecular, Facultad de Ciencias Químico-Biológicas, Universidad Autónoma de Guerrero, Chilpancingo 39090, Guerrero, Mexico; brenddc@hotmail.com (B.D.l.C.-C.); billades@uagro.mx (B.I.-A.); zak_ana@yahoo.com.mx (A.E.Z.-G.); trisrael5@yahoo.com.mx (F.I.T.-R.); 2Laboratorio de Biología Molecular y Genómica, Facultad de Ciencias Químico-Biológicas, Universidad Autónoma de Guerrero, Chilpancingo 39090, Guerrero, Mexico; moniespinoza@yahoo.com; 3Laboratorio de Toxicología, CONACYT-Universidad Autónoma de Guerrero, Chilpancingo 39090, Guerrero, Mexico; paty_fitz@hotmail.com; 4Departamento de Química y Bioquímica, Tecnológico Nacional de México, Instituto Tecnológico/IT de Zacatepec, Calzada Tecnológico 27, Centro, Zacatepec 62780, Morelos, Mexico; macdiel.acevedoqui@uaem.mx; 5Laboratorio de Biología Celular del Cáncer, Facultad de Ciencias Químico-Biológicas, Universidad Autónoma de Guerrero, Chilpancingo 39090, Guerrero, Mexico; nnavarro@uagro.mx; 6Laboratorio de Investigación en Biomoléculas, Facultad de Ciencias Químico-Biológicas, Universidad Autónoma de Guerrero, Chilpancingo 39090, Guerrero, Mexico; wences2009@hotmail.com

**Keywords:** *Ficus crocata*, Moraceae, antioxidants, hydrogen peroxide, cytoprotective effect, antitumor activity, cervical cancer

## Abstract

Oxidative stress causes several chronic diseases including cancer. Some chemotherapeutic agents are not selective against tumor cells, causing oxidative stress in non-tumor cells. This study aimed to evaluate the cytotoxic effect of acetone extract of *Ficus crocata*
*(Miq.) Mart. ex Miq. (F. crocata)* leaves (Ace-EFc) on cervical cancer cells, as well as its protective effect on hydrogen peroxide (H_2_O_2_)-induced lipoperoxidation and cytotoxicity in non-tumor HaCaT cells. Antioxidant activity was determined using the DPPH and ABTS radicals. Cell viability and lipoperoxidation were determined with MTT and 1-methyl-2-phenylindole assays, respectively. A model of H_2_O_2_-induced cytotoxicity and oxidative damage in HaCaT cells was established. HaCaT cells were exposed to the extract before or after exposure to H_2_O_2_, and oxidative damage and cell viability were evaluated. Ace-EFc inhibited the DPPH and ABTS radicals and showed a cytotoxic effect on SiHa and HeLa cells. Furthermore, the extract treatment had a protective effect on hydrogen peroxide-induced lipoperoxidation and cytotoxicity, avoiding the increase in MalonDiAldehyde (MDA) levels and the decrease in cell viability (*p* < 0.001). These results suggest that the metabolites of *F. crocata* leaves possess antioxidant and cytoprotective activity against oxidative damage. Thus, they could be useful for protecting cells from conditions that cause oxidative stress.

## 1. Introduction

The introduction oxidative stress is an imbalance of the redox state generated by the high production of reactive oxygen species (ROS) and the low antioxidant capacity of cells [1]. ROS, such as hydrogen peroxide (H_2_O_2_), are constantly produced as a by-product of metabolism and play an important role in cellular homeostasis [2]. However, there are physiological and environmental factors, such as inflammation and exposure to ultraviolet (UV) radiation, that increase their cellular concentrations [3,4]. ROS can react with biomolecules and damage their structure and function [5], as well as dysregulate cell metabolism, proliferation, differentiation, and survival in tissues [6].

The overproduction of free radicals is known to cause several chronic degenerative diseases including cancer [7]. Although cells possess antioxidant systems to prevent oxidative damage when there is oxidative stress, the antioxidant system is usually deficient in maintaining ROS levels below the risk threshold. Consequently, the exogenous antioxidants available in food or plants are required to maintain the redox balance in the cell [1,8]. Pharmacological studies have shown that some species of the *Ficus* genus (Moraceae) exert antioxidant and protective activity against oxidative damage [9,10,11], and their biological effects have been related to the antioxidant activity of the secondary metabolites present in the plant [12]. Many *Ficus* species remain scarcely studied, such as *Ficus crocata (Miq.) Mart. ex Miq. (F. crocata)*, one of the most widely distributed *Ficus* species in Mexico, is used in infusions (bark and leaves) as a traditional medicine for the control of some diseases, such as diabetes and hypertension [13].

The chemotherapeutic agents used in conventional cancer therapy are not selective against tumor cells, causing oxidative stress in non-tumor cells and therefore side effects [14], so the search for selective cytotoxic therapeutics against tumor cells continues. Previously, we reported that the dichloromethane and acetone extract of *F. crocata* leaves decreased the proliferation capacity of MDA-MB-231 triple-negative breast cancer cells [15]. In this regard, and interestingly, several of the compounds identified in the extracts have been reported as antioxidants; however, their antioxidant activity has not, to our knowledge, been analyzed. This study aimed to evaluate the cytotoxic effect of the acetone extract of *F. crocata* leaves (Ace-EFc) on cervical cancer SiHa and HeLa cells, as well as the antioxidant activity and the protective effect of the extract on hydrogen peroxide (H_2_O_2_)-induced lipoperoxidation in non-tumor HaCaT cells.

## 2. Results

### 2.1. Ace-EFc Has Antiradical Activity Comparable or Superior to That of Ascorbic Acid

GC-MS analysis showed the presence of diterpenes, triterpenes, sterols, and tocopherol in the Ace-EFc. Based on abundance, α-tocopherol (42.4%) and squalene (24.0%) were the major compounds in the extract, followed by stigmastan-3,5-diene (9%), lupeol (8.6%), phytol (8.5%), and β-sitosterol (7.5%) (Table 1) (Appendix A). Interestingly, Ace-EFc inhibited the DPPH^•^ and ABTS^•+^ radicals, demonstrating inhibition activity in both assays compared with the standard ascorbic acid (Appendix A). However, activity was higher on ABTS^•+^ (IC_50_ of Ace-EFc = 1.47 ± 1.21 μg/mL, and IC_50_ of AA = 2.22 ± 0.80 μg/mL, *p* < 0.05) than on the DPPH^•^ radical (IC_50_ of EAFC = 107.05 ± 2.61 μg/mL, and IC_50_ of AA = 118.82 ± 2.48 μg/mL, *p* < 0.05) (Table 2).

### 2.2. Ace-EFc Is Cytotoxic on SiHa and HeLa Cervical Cancer Cells but Not in HaCaT Non-Tumor Cells

HaCaT, SiHa, and HeLa cells were exposed to different Ace-EFc concentrations (0–320 μg/mL) and cell viability was determined. Exposure to Ace-EFc induced a concentration-dependent cytotoxic effect (<80% viable cells) from 40 to 320 μg/mL and from 5 to 40 μg/mL in SiHa and HeLa cells, respectively (Figure 1); In HeLa cells, the percentage of viable cells recovered at concentrations greater than 80 μg/mL, however, the number of viable cells remained below 80%. In contrast, in HaCaT cells, no changes were observed compared with the control at 1.25–20 µg/mL, and the Ace-EFc concentrations of 40–320 μg/mL decreased cell viability compared to control (*p* < 0.001). Nevertheless, the percent of viable cells remained above 80% under all conditions. Thus, it is concluded that Ace-EFc does not exert a cytotoxic effect on HaCaT cells (Figure 1). Ace-EFc showed a greater cytotoxic effect on SiHa cells (IC_50_ = 196.99 ± 2.70 μg/mL, *p* < 0.001) followed by HeLa cells (IC_50_ = 463.30 ± 3.23 μg/mL, *p* < 0.01) compared to HaCaT cells (IC_50_ = 734.33 ± 2.20 μg/mL) (Table 3). Considering these observations, the concentrations of 1.25–20 μg/mL were considered to evaluate the antioxidant activity of Ace-EFc on HaCaT cells.

HaCaT cells were exposed to different concentrations (200–1000 µM) and exposure times (0.5–4 h) of H_2_O_2_ to establish the concentrations that induce cytotoxicity and oxidative damage in this cell model. Cytotoxicity and increased lipoperoxidation were observed at 1000 µM H_2_O_2_ for 2 h; under these conditions, MDA levels increased 700% compared to the control (*p* < 0.0001) (Figure 2a), and the percent of viable cells was reduced to 69% (*p* < 0.0001) (Figure 2b).

### 2.3. Ace-EFc Protects HaCaT Cells from H_2_O_2_-Induced Lipoperoxidation and Cytotoxicity

To assess the preventive effect of Ace-EFc on H_2_O_2_-induced oxidative damage, HaCaT cells were pretreated with Ace-EFc for 24 h and were subsequently treated with 1000 µM H_2_O_2_ for 2 h. Interestingly, Ace-EFc pretreatment prevented lipoperoxidation in a concentration-dependent manner compared to cells treated with H_2_O_2_ alone (*p* < 0.0001). Concentrations of 1.25, 2.5, 5, 10, and 20 µg/mL of EAFC reduced the formation of MDA 9.9, 7.3, 15.2, 25.1, and 143.5 times, respectively, compared to the group treated with H_2_O_2_, which increased MDA formation 8.2 times compared to the control (*p* < 0.0001) (Figure 3). No statistical differences at the MDA level were observed in the cells treated with Ace-EFc compared to the untreated cells. However, a tendency to decrease in a concentration-dependent manner from 5 µg/mL was observed (Figure 3).

To assess the preventive effect of Ace-EFc on H_2_O_2_-induced cytotoxicity, HaCaT cells were pretreated with Ace-EFc for 24 h and subsequently were treated with 1000 µM H_2_O_2_ for 2 h. Ace-EFc exhibited a protective effect against H_2_O_2_-induced cytotoxicity. Previous exposure to concentrations of 1.25, 2.5, 5, 10, and 20 µg/mL of Ace-EFc maintained cell viability at 94.9%, 93.7%, 94.9%, 92.7%, and 83.1%, respectively, even after H_2_O_2_ treatment, whereas when cells were treated with H_2_O_2_ alone, 69% of viable cells were observed (*p* < 0.001) (Figure 4a). Moreover, H_2_O_2_ treatment induced morphological cell changes, such as apparent damage to the cell membrane and a reduction in size and vesiculation in some cells, which could be associated with cell death. However, pretreatment with Ace-EFc avoided these cellular changes, and even cells treated with H_2_O_2_ retained a similar morphology to untreated cells (Figure 4b).

Taken together, these observations suggest that the compounds present in Ace-EFc have a selective cytotoxic effect against cervical tumor cells, without affecting non-tumor cells, in which the compounds could behave as antioxidants, even providing a cytoprotective effect against oxidative molecules.

## 3. Discussion

The overproduction of free radicals is known to cause several chronic degenerative diseases including cancer; however, some therapeutic molecules increase the production of free radicals to induce the death of tumor cells [7,16]. Moreover, the chemotherapeutic agents used in conventional cancer therapy are not selective against tumor cells, causing oxidative stress in non-tumor cells and therefore side effects affecting the health of patients [14]. So, it is necessary to find compounds with selective cytotoxic activity against tumor cells. In this regard, some compounds such as curcumin, have been reported to show a prooxidant effect in tumor cells inducing cell death, while they show an antioxidant effect in non-tumor cells [16]. Here, we reported the cytotoxic effect of acetone extract of *Ficus crocata* on cervical tumor SiHa and HeLa cells (Figure 1), while in non-tumor cells HaCaT the extract showed a protective effect against H_2_O_2_-induced lipoperoxidation and cytotoxicity (Figure 3 and Figure 4). Considering the above, the compounds present in Ace-EFc act as antioxidants in the HaCaT non-tumor cells and probably they have a prooxidant effect in the HeLa and SiHa cervical cancer cells.

We observed that Ace-EFc inhibited DPPH^•^ and ABTS^•+^ radicals, however, the activity was higher on ABTS^•+^ than on the DPPH^•^ radical (Table 2). It is reported that some extracts with inhibitory activity on the ABTS^•+^ radical did not exhibit inhibitory activity on the DPPH^•^ radical, due to the fact that ABTS^•+^ reacts energetically with electrons and has less selectivity in the reaction with atom hydrogen donors [17]. In contrast, the DPPH^•^ radical is more selective than ABTS^•+^ in the reaction with hydrogen donors and does not react with aromatic acids containing only one OH group [18]. This suggests that the antioxidant activity of Ace-EFc by inhibiting DPPH^•^ and ABTS^•+^ radicals is due to the presence and synergistic activity of compounds that participate in both the donation of hydrogen atoms and electrons, such as tocopherols, triterpenes, diterpenes, and phytosterols, which are recognized as antioxidants because of their ability to capture free radicals [5,19,20].

To evaluate the antioxidant activity of Ace-EFc, we first established a model of H_2_O_2_-induced cytotoxicity and oxidative damage in HaCaT cells; cytotoxicity and increased lipoperoxidation were observed at 1000 µM H_2_O_2_ for 2 h (Figure 2). Apparently, H_2_O_2_ was converted into OH^•^ radicals in the presence of Fe^2+^ through the Fenton reaction within the cell [21], altering intracellular macromolecules such as the polyunsaturated fatty acids of the membranes, in that they are the main targets of ROS, causing lipid peroxidation [22]. This could explain the induction of lipoperoxidation at a short exposure time with H_2_O_2_ (2 h) observed in Figure 2a, and, also in terms of causing lipoperoxidation, high concentrations of ROS can damage other macromolecules such as DNA and proteins, giving rise to oxidative damage in organelles and consequently, cell death [5,23].

The cytotoxic effect of H_2_O_2_ on HaCaT cells was in agreement with that previously reported, and the reduction in cell viability corresponds to the oxidative damage caused by the increase in H_2_O_2_ concentration [20,23,24,25]. Considering that mitochondrial homeostasis plays a key role in cell viability and survival [26], the cytotoxic effect has been associated with mitochondrial dysfunction and H_2_O_2_-induced apoptotic events [20,23,24,25]. Properly functioning mitochondria produce ATP and modulate cellular redox balance to ensure cell metabolism [27,28]. However, damaged mitochondria participate in apoptotic signal activation and amplification through multiple effects [29]. H_2_O_2_-induced oxidative damage in HaCaT cells has been reported to induce mitochondrial dysfunction by increasing concentrations of ROS, decreasing antioxidant defenses (SOD, GPx, and GSH), and increasing the release of cytochrome C into the cytoplasm and nucleus [24]. On the other hand, exposure to H_2_O_2_ is known to promote apoptosis, because it increases the expression and activation of pro-apoptotic proteins, such as Bax and caspases 3, 6, 7, 8, and 9, and reduces the expression of the anti-apoptotic protein Bcl-2 [21,24,25]. These events could explain the H_2_O_2_-induced oxidative damage and cytotoxicity in HaCaT cells.

To assess the preventive effect of Ace-EFc on H_2_O_2_-induced oxidative damage, HaCaT cells were pretreated with Ace-EFc for 24 h and were subsequently treated with H_2_O_2_. We observed that Ace-EFc pretreatment prevented H_2_O_2_-induced lipoperoxidation in a concentration-dependent manner. The compounds identified in the Ace-EFc (diterpenes, triterpenes, sterols, and tocopherols) have been recognized as antioxidants due to their ability to scavenge free radicals. Thus, they are capable of regulating their concentration, preventing the morphological alterations and oxidative damage caused by oxidative stress inducers [5,19,20]. Triterpenes and diterpenes have been shown to sequester and decrease OH^•^ production, respectively. Triterpenes have greater reducing power than endogenous glutathione (GSH) and possess strong uptake activity against O_2_^•−^ and its metabolites [30]. On the other hand, phytosterols and tocopherols reduce H_2_O_2_ through the donation of a hydrogen atom [19,24], which demonstrates the high antioxidant capacity of these phytochemicals in reducing free radicals. It is suggested that the compounds present in the Ace-EFc regulated ROS concentrations in the cells and potentiated the expression and activity of endogenous antioxidants. This is because tocopherols, triterpenes, diterpenes, and phytosterols have been demonstrated to regulate the concentration and activity of GSH, GPx, GSTM1, GR, SOD, and CAT. Moreover, these compounds prevent oxidative damage in macromolecules, preventing lipoperoxidation, therefore reducing MDA levels [5,31,32,33,34], which is consistent with the results in the present study. α-Tocopherol was the major compound identified in the Ace-EFc (Table 1). Tocopherols are important antioxidants for cell protection. The presence of these compounds in the cell can influence the decrease in oxidative stress, thus preventing the oxidation of polyunsaturated fatty acids in cell membranes [35,36]. The activity of tocopherols has been attributed to their lipophilic nature, which facilitates their free distribution in and action on the membrane, favoring the elimination of lipoperoxides [37], which could explain the reduction in MDA when the cells were treated with Ace-EFc.

On the other hand, we observed that Ace-EFc exhibited a protective effect against H_2_O_2_-induced cytotoxicity on HaCaT cells. These results are consistent with several studies in which the cytoprotective activity of the compounds identified in Ace-EFc was reported against the toxicity of oxidative stress inducers. It is reported that tocopherols and triterpenes possess neuroprotective [20,38], hepatoprotective [5,34], cardioprotective [39], and cytoprotective activity on sperm [40] and testicular morphological alterations [33] in rats and mice. Specifically, α-tocopherol showed to reduce ROS levels in HaCaT cells exposed to UV light, thereby increasing the cell viability altered by this oxidative stress inducer [41]. Furthermore, tocopherols increase the survival rate in renal epithelial cells exposed to oxidative stress, as they prevent the depolarization of the mitochondrial membrane by decreasing H_2_O_2_ levels, also decreasing caspase-3 levels [42]. Triterpenes, in addition to preventing lipoperoxidation, avoid protein carbonylation, and protect the permeability of the mitochondrial membrane by avoiding the decrease of Bcl-2 and the increase of Bax, important regulators of the apoptotic cascade. Triterpenes also decrease the release of cytochrome C and the level of caspases 9/3 and, inhibit DNA damage by reducing the formation of 8-OH-dG and DNA fragmentation [5,34,43]. On the other hand, exposure to tocopherols has also resulted in significant protection against damage to the cell membrane [35], which is consistent with the observation that Ace-EFc prevents H_2_O_2_-caused damage on the membrane and cell morphology.

Taken together, the data suggest that the compounds present in Ace-EFc possess antioxidant activity and protect cells from oxidative damage, regulating the antioxidant system of cells and trapping the free radicals generated by the inducers of oxidative stress. However, further studies are required to delve deeper into the mechanism of action that Ace-EFc compounds employ to protect cells from the different inducers of oxidative damage. This study provides, to our knowledge for the first time, information on the antioxidant capacity of the leaf extracts of *F. crocata* in a non-tumor cell model. Previously, we demonstrated that these extracts revealed antiproliferative activity in MDA-MB-231 breast cancer cells, inducing cell cycle arrest and apoptosis [15], which highlights the importance of continuing to study the effect of these compounds, their potential use as antioxidants in non-tumor cells, and as antiproliferative agents in tumor cells.

In conclusion, the acetone leaf extract of *F. crocata* showed selective cytotoxic activity against SiHa and HeLa cervical cancer cells, without affecting HaCaT non-tumor cells. More studies are required to analyze the mechanisms by which the compounds identified in the extract induce cytotoxicity in cervical tumor cells. Besides, the acetone leaf extract of *F. crocata* has antioxidant activity comparable or superior to that of ascorbic acid, due to its containing compounds with antioxidant activity, such as α-tocopherol, squalene, stigmastan-3,5-diene, lupeol, phytol, and β-sitosterol, in order of abundance, and prevents H_2_O_2_-induced lipoperoxidation and cytotoxicity in HaCaT cells.

## 4. Materials and Methods

### 4.1. Plant Material and Preparation of the Extract

Leaves of *Ficus crocata (Miq.) Mart. ex Miq.* were collected from Petaquillas, Guerrero, Mexico (17°29′15″ N and 99°27′35″ W) and authenticated in agreement with that previously described by Sánchez-Valdeolivar et al. [15], a specimen of the plant is deposited at Herbario Nacional de México (MEXU), voucher number MEXU:1100067. The samples (100 g) were cleaned, dried for over 8 days at 39 °C protected from light, and then pulverized. The powder was macerated with acetone solvent (reactive-grade, 500 mL, during 24 h, three times). The macerated material was filtered, and the organic phase was evaporated in a rotary evaporator (Digital Rotary Evaporator Model 410, Puebla, México) at 60 °C and 80 rpm. The acetone extracts were stored at −20 °C and protected from light until their use.

### 4.2. Phytochemical Profile of Ace-EFc

The volatile compounds present in the Ace-EFc were identified by Gas Chromatography-Mass spectrometry (GC-MS) analysis. The analyses were carried out in triplicate in an Agilent 6890 series Gas Chromatograph equipped with a mass selective detector (5973N, Santa Clara, CA, USA). The experimental conditions of the GC-MS system were the following: HP-5MS capillary nonpolar column (30 m; ID: 0.20 mm; film thickness: 0.25 μm). The carrier gas was helium at a flow rate of 1.0 mL/min. Concerning the gas chromatography, the temperature program (oven temperature) was 50 °C, raised to 230 °C at 2 °C/min, and the injection volume was 1 μL [15]. All results were compared by utilizing the NIST/EPA/NIH Mass Spectral library version 1.7a (ChemStation, Santa Clara, CA, USA).

### 4.3. DPPH^•^ and ABTS^•+^ Radical Inhibition Assays

To evaluate the antiradical activity of Ace-EFc, 0.05–200 µg/mL of Ace-EFc was placed in 96-well plates (Corning, NY, USA), and 150 µL of 0.3 mM 2,2-DiOhenyl-1-PicrylHydrazyl (DPPH^•^) or 2,2′-AzinoBis-(3-ethylbenzoThiazoline-6-Sulfonic acid) (ABTS^•+^) radicals were added [44,45]. Previously, the ABTS^•+^ reagent was produced by reacting the ABTS solution (7 mM) with potassium persulfate [2.45 mM]. After incubation for 16 h at room temperature, 1 mL of this solution was diluted in 60 mL of methanol. In both tests, methanol was employed as blank, while ascorbic acid (0.05–200 µg/mL) was used as standard antioxidant. The reactions between Ace-EFc and DPPH^•^ or ABTS^•+^ were incubated in the dark at room temperature for 30 min, then read at 545 nm (Star Fax 2100, Awareness Technologies, Palm city, FL, USA) and 734 nm (Multiskan FC Plate Leader, LabSystems, Waltham, MA, USA), respectively. The half-maximal Inhibitory Concentration (IC_50_) was calculated through the linear equation (Y = mX + b) using GraphPad Prism v6.0 software. The percentage of radical inhibition was determined by the following formula:(1)$$\%\;\mathrm{inhibition}\;=\;(1-\left(\frac{Ab_{}\;\mathrm{Sample}}{Ab_{}\;\mathrm{Blank}}\right)\;\times \;100$$

### 4.4. Cell Culture and H_2_O_2_-Induced Lipoperoxidation and Cytotoxicity Model

HaCaT, SiHa, and HeLa cell lines were obtained and previously certified from *Instituto Nacional de Cancerología* (CDMX, Mexico). HaCaT, SiHa, and HeLa cells were cultured in Dulbecco’s Modified Eagle’s Medium formula 12 (DMEM/F12) supplemented with 10% Fetal Bovine Serum (FBS), 1% antibiotic (Ampicillin/Streptomycin), and incubated at 37 °C in a 5% CO_2_ atmosphere at 95% humidity and were grown to 80% confluence. To establish H_2_O_2_ concentrations inducing cytotoxicity and oxidative damage, the HaCaT cells were exposed to 0, 200, 400, 600, 800, and 1000 µM of H_2_O_2_ during 0.5, 1, 2, and 4 h, and cell viability and lipoperoxidation were determined.

### 4.5. Cell Viability Assays

Cell viability was evaluated using the MTT [3-(4,5-dimethylthiazol-2-yl)-2,5-diphenyltetrazolium bromide] cell proliferation colorimetric assay (CT02, Millipore Corp., Bedford, MA, USA) according to the manufacturer’s instructions. Briefly, in a 96-well plate, 1 × 10^4^ HaCaT, SiHa, or HeLa cells per well were cultured with DMEM/F12 medium with 10% FBS for 24 h. The cells were treated with 0–320 µg/mL of Ace-EFc for 24 h. After the treatment, the medium containing the extract was replaced by fresh basal medium and 100 µL of the MTT reagent was added for 4 h. The formazan crystals were diluted with Isopropanol, and the Optical Density (OD) of the supernatant was obtained at a wavelength of 545/630 nm (Star Fax 2100, Awareness Technologies, Palm city, FL, USA). To assess the protective effect of Ace-EFc on H_2_O_2_-induced cytotoxicity, HaCaT cells were initially treated with Ace-EFc as described previously. The medium was discarded, and subsequently, cells were treated with H_2_O_2_ (1000 µM for 2h) according to the model established H_2_O_2_-induced cytotoxicity. The percentage of viable cells was determined by the following formula:(2)$$\mathrm{Viable}\;\mathrm{cells}\;=\;\left(\frac{Ab_{}\;\mathrm{Sample}}{Ab_{}\;\mathrm{Blank}}\right)\;\times \;100$$

### 4.6. Oxidative Damage Determination (Lipoperoxidation)

HaCaT cells were grown in a six-well plate to 80% confluence and were treated with 0–20 μg/mL of Ace-EFc for 24 h. Later, the medium was discarded, and subsequently, cells were treated with H_2_O_2_ (1000 µM for 2h), according to the model established H_2_O_2_-induced cytotoxicity. The supernatant was recovered (2 mL), and the cells were trypsinized and resuspended in 65 µL PBS (pH 7.4), and vortexed for 1 min. To quantify MalonDiAldehyde (MDA) levels, 50 µL of the sample (supernatant or cells separately), 50 µL Milli Q water, 350 µL MPI (1-Methyl-2-OhenylIndole) reagent (10 mM), and 100 µL HCl (37%) were mixed, shaken, and they were incubated for 40 min at 45 °C. They were subsequently centrifuged at 4 °C at 7000 rpm for 10 min and the absorbance was read at 586 nm (NanoDrop 2000, Waltham, MA, USA) [46]. The data were normalized with the total protein concentration of each sample obtained by the Bradford assay. The final concentration of MDA in each sample was obtained by adding the data obtained from both the supernatant and the cell suspension.

### 4.7. Statistical Analysis

Data analysis was performed using GraphPad Prism version 6.0 statistical software. The data were shown as the mean ± standard deviation (SD). One-way analysis of variance (ANOVA) was used with the Dunnett multiple comparison test. Statistically significant differences were considered when *p* < 0.05.

## Figures and Tables

**Figure 1 plants-10-00183-f001:**
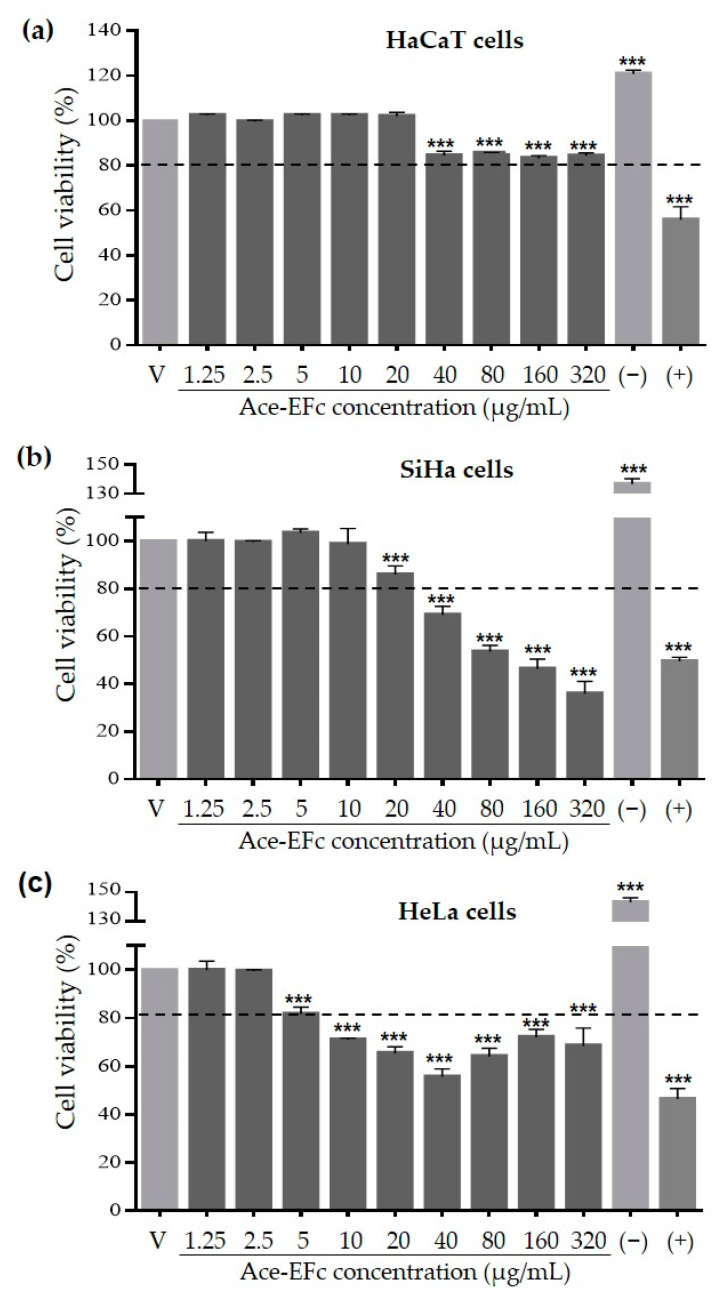
Effect of acetone extract from *F. crocata* (Ace-EFc) on cell viability. HaCaT, SiHa, and HeLa cells were treated with 0–320 µg/mL of Ace-EFc for 24 h. MTT assay for (**a**) HaCaT, (**b**) SiHa, and (**c**) HeLa cells. V: vehicle, cells treated with diluent of the extracts (DMSO < 1%); (−): negative control, 10% FBS. (+): Positive control, 100 μM Ara-C (cytarabine). Results were expressed as the mean ± SD of three independent experiments. One-way ANOVA, Dunnett’s test: *** *p* < 0.001 versus C.

**Figure 2 plants-10-00183-f002:**
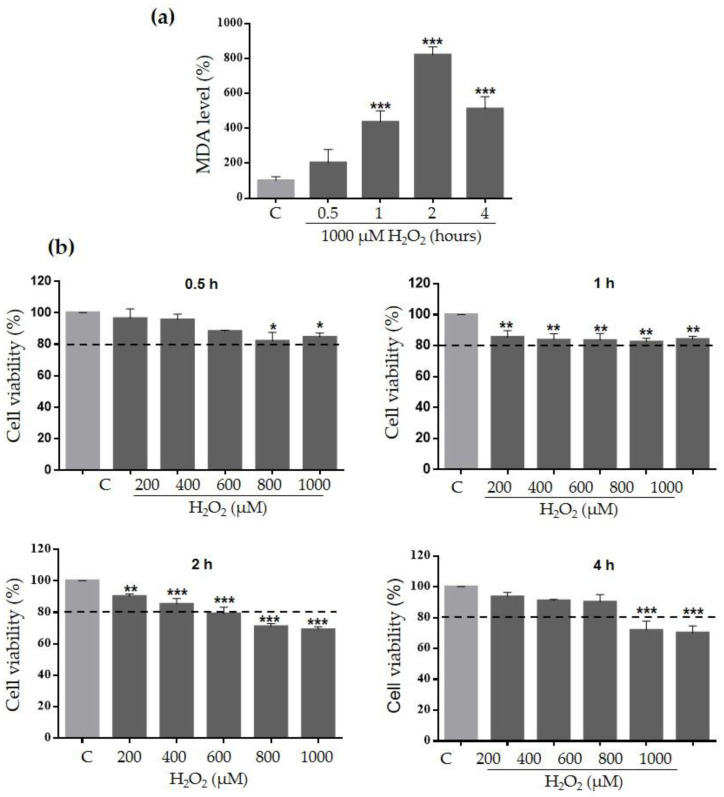
Establishment of H_2_O_2_-induced oxidative damage and cytotoxicity model on HaCaT cells. (**a**) Quantification of MDA levels to the determination of H_2_O_2_-induced lipoperoxidation. HaCaT cells were treated with 1000 µM H_2_O_2_ for 0.5, 1, 2, or 4 h. (**b**) MTT assay to the determination of H_2_O_2_-induced cytotoxicity. HaCaT cells were treated with H_2_O_2_ (200–1000 µM) for 0.5, 1, 2, or 4 h. Results were expressed as the mean ± SD of three independent experiments. One-way ANOVA, Dunnett’s test: * *p* < 0.05, ** *p* < 0.01, *** *p* < 0.001 versus C.

**Figure 3 plants-10-00183-f003:**
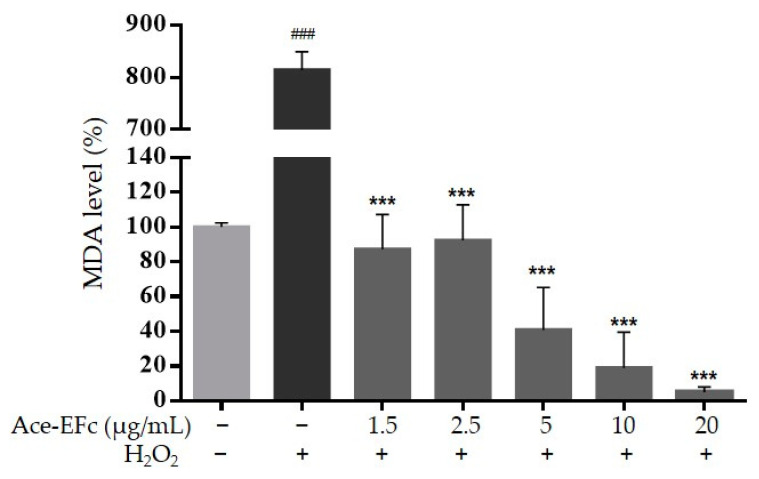
Protective effect of Ace-EFc against H_2_O_2_-induced lipoperoxidation. HaCaT cells were pretreated with Ace-EFc (1.25, 2.5, 5, 10, and 20 µg/mL) for 24 h, and later, treated with 1000 µM H_2_O_2_ for 2 h. Results were expressed as the mean ± SD of three independent experiments. One-way ANOVA, Dunnett’s test: ^###^
*p* < 0.001 versus no treated cells, *** *p* < 0.001 versus treated cells with H_2_O_2_.

**Figure 4 plants-10-00183-f004:**
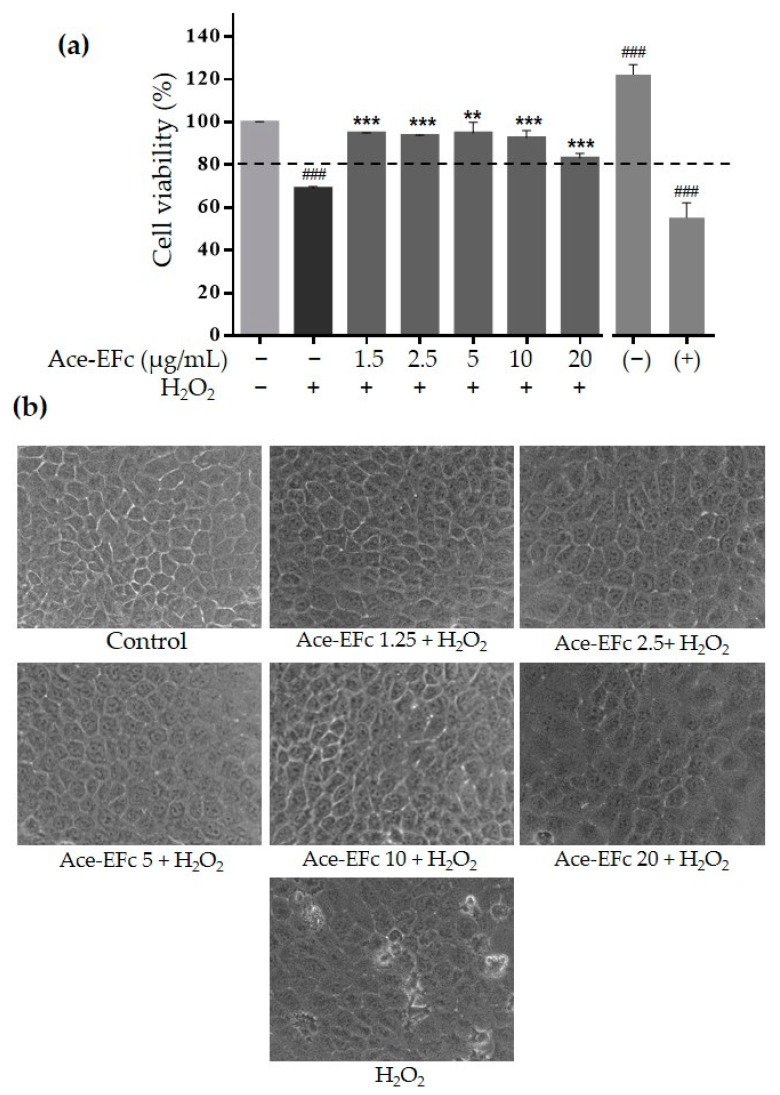
Protective effect of Ace-EFc against H_2_O_2_-induced cytotoxicity. HaCaT cells were pretreated with Ace-EFc (1.25, 2.5, 5, 10, and 20 µg/mL) for 24 h before treatment with 1000 µM H_2_O_2_ for 2 h. (**a**) MTT assays. (−): negative control, cells stimulated with 10% FBS. (+). Positive control, 100μM Ara-C (cytarabine). Results were expressed as the mean ± SD of three independent experiments. One-way ANOVA, Dunnett’s test: ^###^
*p* < 0.001 versus no treated cells; ** *p* < 0.01, *** *p* < 0.001 versus treated cells with H_2_O_2_. (**b**) Micrographs of HaCaT cells from (**a**).

**Table 1 plants-10-00183-t001:** The chemical composition of acetone extract from *F. crocata* (Ace-EFc).

No.	Compound	PubChem CID	Family	Molecular Formula	Molecular Weight	Abundance (%) *
1	Phytol	5366244	Diterpene	C_20_H_40_O	296.539	8.50
2	Squalene	638072	Triterpene	C_30_H_50_	410.73	24.00
3	Stigmastan-3,5-diene	525918	Sterol	C_29_H_48_	396.703	9.00
4	Alpha-tocopherol	14985	Tocopherol	C_29_H_50_O_2_	430.717	42.40
5	β-sitosterol	222284	Sterol	C_29_H_50_O	414.718	7.50
6	Lupeol	259846	Triterpene	C_30_H_50_O	426.729	8.60

* Percentage of the total, considering the compounds identified in GC-MS analysis.

**Table 2 plants-10-00183-t002:** IC_50_ for DPPH and ABTS free radical scavenging activity of Ace-EFc.

Samples	IC_50_ (μg/mL) ± SD	
	DPPH	ABTS
Ace-EFc	107.05 ± 2.61 *	1.47 ± 1.21 *
Ascorbic Acid	118.82 ± 2.48	2.22 ± 0.80

Samples were analyzed using *t* test: * *p* < 0.05 versus ascorbic acid. DPPH = 2,2-Diphenyl-1-picrylhydrazyl; ABTS = 2,2′-azinobis-(3-ethylbenzothiazoline-6-sulfonic acid); SD = standard deviation.

**Table 3 plants-10-00183-t003:** IC_50_ of Ace-EFc on the viability of HaCaT, SiHa, and HeLa cells.

Cell Line	IC_50_ (μg/mL) ± SD
HaCaT	734.33 ± 2.20
SiHa	196.99 ± 2.70 ***
HeLa	463.30 ± 3.23 **

Samples were analyzed using *t* test. ** *p* < 0.01, *** *p* < 0.001 compared to HaCaT cells. Ace-EFc = acetone extract from *Ficus crocata*, SD = standard deviation.

## Data Availability

The data presented in this study are available within the article and its Appendix A.

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
