# Peer review of "Cytotoxicity of Ficus Crocata Extract on Cervical Cancer Cells and Protective Effect against Hydrogen Peroxide-Induced Oxidative Stress in HaCaT Non-Tumor Cells"

_plants, 2021, doi:10.3390/plants10010183_

Round 1
Reviewer 1 Report
The results are interesting, but cannot accepted in the present form on the base of these observations:
-The positive control should be reported in the different assays
- The study could be acquire more value by testing some parameters of cell death
The study reported the cytotoxicity of Ficus crocata extract on cervical cancer cells and protective effect against hydrogen peroxide-induced oxidative stress in HaCaT non-tumor cells. Interestingly, the acetone leaf extract of F. crocata showed selective cytotoxic activity against SiHa and HeLa cervical cancer cells, without affecting HaCaT non-tumor cells. In addition, the acetone leaf extract of F. crocata has antioxidant activity comparable or superior to that of ascorbic acid, due to its containing compounds with antioxidant activity, such as α-tocopherol, squalene, stigmastan-3,5-diene, lupeol, phytol, and β-sitosterol, and prevents H2O2-induced lipoperoxidation and cytotoxicity in Ha-266 CaT cells. However, the manuscript cannot accepted in the present form on the base of these observations:
-Authors (page 6, lines: 139-141) reported “Moreover, H2O2 treatment induced morphological cell changes, such as apparent damage to the cell membrane and a reduction in size and vesiculation in some cells, which could be associated with cell death.” Some parameters of cell death, for example DNA damage (DNA fragmentation analysis) and cell membrane permeability by Lactate dehydrogenase (LDH) release should be carried out.
-The positive control should be reported in the “Effect of Ace-EFc on cell viability” and “Protective effect of Ace-EFc against H2O2-induced lipoperoxidation” (MTT and 1-methyl-2-26 phenylindole assays)
Author Response
Response to Reviewer 1 Comments
REVIEWER COMMENT. The study reported the cytotoxicity of Ficus crocata extract on cervical cancer cells and protective effect against hydrogen peroxide-induced oxidative stress in HaCaT non-tumor cells. Interestingly, the acetone leaf extract of F. crocata showed selective cytotoxic activity against SiHa and HeLa cervical cancer cells, without affecting HaCaT non-tumor cells. In addition, the acetone leaf extract of F. crocata has antioxidant activity comparable or superior to that of ascorbic acid, due to its containing compounds with antioxidant activity, such as α-tocopherol, squalene, stigmastan-3,5-diene, lupeol, phytol, and β-sitosterol, and prevents H2O2-induced lipoperoxidation and cytotoxicity in Ha-266 CaT cells. However, the manuscript cannot accepted in the present form on the base of these observations:
POINT 1. The positive control should be reported in the different assays. The positive control should be reported in the “Effect of Ace-EFc on cell viability” and “Protective effect of Ace-EFc against H2O2-induced lipoperoxidation” (MTT and 1-methyl-2-26 phenylindole assays).
RESPONSE: Thank you for the time inverted in the revision of this manuscript. The positive and negative controls for cell proliferation (cell viability) were added to Figures 1 and 4. The cell viability assays now show three controls: Vehicle, diluent of the extracts (DMSO 0.32%); Negative control (-), cells treated with FBS 10% to induce cell proliferation; and Positive control (+), cells treated with 100μM Ara-C (cytarabine) which is known to induce cell death under these conditions.
- Hewish, Martin SA, et al., 2013. Cytosine-based nucleoside analogs are selectively lethal to DNA mismatch repair deficient tumour cells by enhancing levels of intracellular oxidative stress. British Journal of Cancer; 108: 983–992. doi: 10.1038/bjc.2013.3.
- Frédéric Dessi et al, 1995. Cytosine Arabinoside Induces Apoptosis in Cerebellar Neurons in Culture. https://doi.org/10.1046/j.1471-4159.1995.64051980.x
In the case of lipoperoxidation assays, there is sufficient evidence that hydrogen peroxide induces oxidative damage in HaCaT cells, so we consider that another positive control is not necessary.
- Yoon JJ, Jeong JW, Choi EO, Kim MJ, Hwang-Bo H, Kim HJ et al (2017) Protective effects of Scutellaria baicalensis Georgi against hydrogen peroxide-induced DNA damage and apoptosis in HaCaT human skin keratinocytes. EXCLI J 16:426-438. https://doi.org/10.17179/excli2016-817
- Chen Z, Wang C, Yu N, Si L, Zhu L, Zeng A et al (2019). INF2 regulates oxidative stress-induced apoptosis in epidermal HaCaT cells by modulating the HIF1 signaling pathway. Biomed Pharmacother 111:151-161. https://doi.org/10.1016/j.biopha.2018.12.046
- Yoon Y, Lee YM, Song S, Lee YY, Yeum KJ (2018). Black soybeans protect human keratinocytes from oxidative stress-induced cell death. Food Sci Nutr 6:2423-2430. https://doi.org/10.1002/fsn3.842
POINT 2. The study could be acquire more value by testing some parameters of cell death. Authors (page 6, lines: 139-141) reported “Moreover, H2O2 treatment induced morphological cell changes, such as apparent damage to the cell membrane and a reduction in size and vesiculation in some cells, which could be associated with cell death.” Some parameters of cell death, for example DNA damage (DNA fragmentation analysis) and cell membrane permeability by Lactate dehydrogenase (LDH) release should be carried out.
RESPONSE: We agree that the manuscript would be more valuable if the cell death parameters were included. However, unfortunately we currently do not have the conditions to make these determinations, in addition to the fact that the main intention of this manuscript is to report the cytotoxic selectivity of the extract against tumor cells without affecting non-tumor cells, as well as the antioxidant and cytoprotective activity of the extract against oxidative damage in non-tumor cells HaCaT, so we do not delve into the mechanisms of cytotoxicity or cell death. In a previous study, we reported that compounds identified in the dichloromethane extract of Ficus crocata leaves induce cell cycle arrest and apoptosis in MDA-MB-231 breast cancer cells (Sánchez-Valdeolivar 2020), so we would expect a similar effect in cervical cancer cells. In subsequent studies we plan to evaluate the biological activity of the pure compounds identified in the extract, and to delve into the mechanisms of cell death, which you mention correctly.
- Sánchez-Valdeolivar, CA, Álvarez-Fitz P, Zacapala-Gómez AE, Acevedo-Quiroz M, Cayetano-Salazár L, Olea-Flores M, et al (2020). Phytochemical profile and antiproliferative effect of Ficus crocata extracts on triple-negative breast cancer cells. BMC Complement Med Ther. https://doi.org/1186/s12906-020-02993-6.
Considering your comment, the sentence “More studies are required to analyze the mechanisms by which the compounds identified in the extract induce cytotoxicity in cervical tumor cells” was added to Discussion section, lines 265-266.
The manuscript was reviewed by a native English speaker (Margaret Ellen Reynolds Adler). In this version, English language and style are fine/minor spell were revised.
Reviewer 2 Report
Please include the correct plant name Ficus crocata (Miq.) Mart. ex Miq (Moraceae)
Please provide the results to assess the identity of each compound.
You need a control drugs for the viability assays.
Author Response
Response to Reviewer 2 Comments
POINT 1. Please include the correct plant name Ficus crocata (Miq.) Mart. ex Miq (Moraceae)
Response: Thank you for the time inverted in the revision of this manuscript. In this version, the name Ficus crocata (Miq.) Mart. ex Miq was used in Abstract, Introduction and Materials and Methods sections (Lines 23, 56 and 272).
POINT 2. Please provide the results to asses the identity of each compound.
Response: In this version, we added the chromatographic profile of the GC-MC analysis as " Figure S1. GC-MS analysis of acetone extract of Ficus crocata (Miq.) Mart. ex Miq. leaves ". This figure shows the chromatogram, structure, and retention time of the compounds identified in the extract.
POINT 3. You need a control drugs for the viability assays.
Response: The positive and negative controls for cell proliferation (cell viability) were added to Figures 1 and 4. The cell viability assays now show three controls: Vehicle, diluent of the extracts (DMSO 0.32%); Negative control (-), cells treated with FBS 10% to induce cell proliferation; and Positive control (+), cells treated with 100μM Ara-C (cytarabine) which is known to induce cell death under these conditions.
- Hewish, Martin SA, et al., 2013. Cytosine-based nucleoside analogs are selectively lethal to DNA mismatch repair deficient tumour cells by enhancing levels of intracellular oxidative stress. British Journal of Cancer; 108: 983–992. doi: 10.1038/bjc.2013.3.
- Frédéric Dessi et al, 1995. Cytosine Arabinoside Induces Apoptosis in Cerebellar Neurons in Culture. https://doi.org/10.1046/j.1471-4159.1995.64051980.
The manuscript was reviewed by a native English speaker (Margaret Ellen Reynolds Adler). In this version, English language and style are fine/minor spell were revised.
Reviewer 3 Report
The authors report the anti-tumor and antioxidant effects of a dry leaf extract of Ficus crocata.
Despite the need for new natural products to protect or reduce the side effects of chemotherapy, this manuscript presents a strong initial perplexity.
The authors use gas chromatography to quantify the secondary metabolites of the dry plant extract. Why do the authors use this technique? Why was the phytochemical profile of secondary metabolites, which should act on cells, done with an ideal technique for volatile components? Furthermore, relative abundance is too generic as a quantification. Do the authors have the ability to quantify with specific Standards?
Authors should show at least the chromatographic profile of their analysis.
I suggest some interesting work to integrate: Gupta AK et al,. Artocarpus lakoocha Roxb. and Artocarpus heterophyllus Lam. Flowers: New Sources of Bioactive Compounds. Plants (Basel). 2020 Oct 9;9(10):1329. doi: 10.3390/plants9101329.
Author Response
Response to Reviewer 3 Comments
REVIEWER COMMENT. The authors report the anti-tumor and antioxidant effects of a dry leaf extract of Ficus crocata. Despite the need for new natural products to protect or reduce the side effects of chemotherapy, this manuscript presents a strong initial perplexity.
POINT 1. The authors use gas chromatography to quantify the secondary metabolites of the dry plant extract. Why do the authors use this technique? Why was the phytochemical profile of secondary metabolites, which should act on cells, done with an ideal technique for volatile components? Furthermore, relative abundance is too generic as a quantification. Do the authors have the ability to quantify with specific Standards?
Response: Thank you for the time inverted in the revision of this manuscript.
The common methods used to identify and quantify the metabolites presents in the extracts are: Gas Chromatography-Mass Spectrometry (GC-MS), Nuclear Magnetic Resonance Spectrometry (NMR), and Liquid Chromatography-Mass Spectrometry (LC-MS). The GC-MS is a common method to identify and quantify the bioactive compounds of the medicinal plants. Also, the mass spectral fingerprint of each compound can be identified from the data library.
Hence, GC-MS is detection technique have become a sophisticated means for analysis of compounds presents in the extracts. We consider that this technique reliably determines the phytochemical profile of the extracts used in this research. This method has been used in several studies to identify compounds and subsequently evaluate their biological activity.
- Ezhilan, B. P., & Neelamegam, R. (2012). GC-MS analysis of phytocomponents in the ethanol extract of Polygonum chinense L. Pharmacognosy research, 4(1), 11–14.
- Ivanov, I., Dincheva, I., Badjakov, I., Petkova, N., Denev, P., & Pavlov, A. (2018). GC-MS analysis of unpolar fraction from ficus carica L. (fig) leaves. International Food Research Journal, 25(1), 282-286.
- El-Beltagi, H., Mohamed, H. I., Abdelazeem, A. S., Youssef, R., & Safwat, G. (2018). GC-MS analysis, antioxidant, antimicrobial and anticancer activities of extracts from ficus sycomorus fruits and leaves. Notulae Botanicae Horti Agrobotanici Cluj-Napoca, 47(2), 493.
- Mujić, I., Bavcon Kralj, M., Jokić, S., Jug, T., Subarić, D., Vidović, S., Jarni, K. (2014). Characterisation of volatiles in dried white varieties figs (ficus carica L.). Journal of Food Science and Technology, 51(9), 1837-1846.
- Imran, M., Rasool, N., Rizwan, K., Zubair, M., Riaz, M., Zia-Ul-Haq, M., . . . Jaafar, H. Z. E. (2014). Chemical composition and biological studies of ficus benjamina.Chemistry Central Journal, 8.
- Swamy, M. K., Sinniah, U. R., & Mohd, S. A. (2015). In vitro pharmacological activities and GC-MS analysis of different solvent extracts of lantana camara leaves collected from tropical region of malaysia. Evidence - Based Complementary and Alternative Medicine, 2015.
The GC-MS is an ideal technique for volatile components, particularly essential oils. However, the current studies focus on the GC-MS use for identification of bioactive compounds in different extracts of medicinal plants, and several author are used the GC-MS to identification of phytochemical profile of polar extracts; by example, Eswaraiah et al (2020) identified the bioactive compounds in methanolic extract of Avicennia alba by GC-MS analysis; Venkateswarulu et al (2020) performed an screening of Ipomoea tuba methanolic extract for identification of bioactive compounds by GC-MS; Al-Nemari et al (2020) used GC-MS to identified the bioactive compounds of methanol, acetone and water extracts from Anona squamosa.
We are aware that the quantification of the compounds using HPLC analysis is necessary. Unfortunately, we do not have the possibility of quantifying the compounds using specific standards. However, we use the "% of the total" parameter reported by the GC-MC analysis, to report the proportion in which the identified compounds were found in the acetone extract. In the caption of Table 1, the sentence “*Percentage of total, considering the compounds identified in GC-MS analysis” was added. In further studies, we plan to evaluate the biological activity using the pure compounds.
POINT 2. Authors should show at least the chromatographic profile of their analysis.
Response: Thank you for the observation. Considering your comment, we added the chromatographic profile of the GC-MC analysis as " Figure S1. GC-MS analysis of acetone extract of Ficus crocata (Miq.) Mart. ex Miq. leaves". This figure shows the chromatogram, structure, and retention time of the compounds identified in the extract through GC-MC.
POINT 3. I suggest some interesting work to integrate: Gupta AK et al,. Artocarpus lakoocha Roxb. and Artocarpus heterophyllus Lam. Flowers: New Sources of Bioactive Compounds. Plants (Basel). 2020 Oct 9;9(10):1329. doi: 10.3390/plants9101329.
Response: Following your suggestion, this reference was added to the manuscript. Reference 8, line 379-381.
The manuscript was reviewed by a native English speaker (Margaret Ellen Reynolds Adler). In this version, English language and style are fine/minor spell were revised.
Round 2
Reviewer 1 Report
The manuscript is acceptable in the present form
Author Response
Thank you for your comments and observations to improve the manuscript.
Reviewer 2 Report
"Many Ficus species remain scarcely studied, such as Ficus crocata (Miq.) Mart. ex Miq. (F. crocata), one of the most widely distributed Ficus species in Mexico. This is used in infusions (bark and leaves) as a traditional medicine for the control of some 58 diseases, such as diabetes and hypertension." X
X = please add the reference, about the traditional use.
I suggest this redaction;
Many Ficus species remain scarcely studied, such as Ficus crocata (Miq.) Mart. ex Miq. (F. crocata), one of the most widely distributed Ficus species in Mexico, is used in infusions (bark and leaves) as a traditional medicine for the control of some diseases, such as diabetes and hypertension.
please review all the manuscript.
Please add the plant voucher number and the herbarium
Author Response
Response to Reviewer 2 Comments
POINT 1. "Many Ficus species remain scarcely studied, such as Ficus crocata (Miq.) Mart. ex Miq. (F. crocata), one of the most widely distributed Ficus species in Mexico. This is used in infusions (bark and leaves) as a traditional medicine for the control of some 58 diseases, such as diabetes and hypertension." X
X = please add the reference, about the traditional use.
I suggest this redaction:
Many Ficus species remain scarcely studied, such as Ficus crocata (Miq.) Mart. ex Miq. (F. crocata), one of the most widely distributed Ficus species in Mexico, is used in infusions (bark and leaves) as a traditional medicine for the control of some diseases, such as diabetes and hypertension. Please review all the manuscript.
Response: Thank you for the time inverted in the revision of this manuscript. We consider the redaction suggested by the reviewer, and the reference was added. Lines 56-59 (Introduction section), and 398-399 (References section).
POINT 2. Please add the plant voucher number and the herbarium.
Response: In Materials and Methods Section, the sentence “… a specimen of the plant is deposited at Herbario Nacional de México (MEXU), voucher number MEXU:1100067” was added. Lines 278-279.
Reviewer 3 Report
The manuscript has improved considerably; the authors have finalized their paper following the indications suggested by the reviewers
I am very satisfied with the corrections and additions made by the authors.
In my opinion, the manuscript can now be published.
Author Response
Response: Thank you for your comments and observations to improve the manuscript.
This manuscript is a resubmission of an earlier submission. The following is a list of the peer review reports and author responses from that submission.